# EDCDE - Extended Discovery of Closed-Form Differential Equations

**Robert Joseph George**[*]
Department of Mathematics and Statistics
University of Alberta
{rjoseph1}@ualberta.ca

## Abstract

Understanding the mathematical connections between variables in a physical system, such as Ordinary Differential Equations (ODEs), is an essential part of the scientific method. This is where symbolic regression plays a key role in looking for closed-form functions given a dataset. We extend the results of the original Discovering Closed-Form ODEs from Observed Trajectories (D-CODE) by considering a generalized variational formulation that can work with most forms of ODEs. We conclude the paper with numerical results and applications.

## 1 Introduction

Symbolic regression for ODEs and PDEs has become a hot topic in recent years due to its significant potential to advance our understanding and modelling of complex systems Simmons (1972). However, the direct application of symbolic regression to ODEs is challenging since we typically cannot observe the time derivative, and the labels for regression are not available in the data. Instead, we only have access to measurements of the state at some discrete time points with added noise Cullum (1971). To overcome this limitation, Qian et al. (2021) studied first-order autonomous systems and observed that finding closed-form ODEs can be a time-consuming and expertise-intensive task. To simplify the analysis or develop efficient numerical algorithms, non-autonomous and higher-order ODEs can be transformed into systems of autonomous ODEs by adding more variables. However, this can be challenging for complex systems and poorly understood equations, requiring significant mathematical analysis and modelling efforts. Additionally, the increase in the number of variables from $n$ to a system of $2n + 1$ can be cumbersome. In our paper, we propose a novel approach to solving original systems of order $n$ using a generalized variational formulation supported by numerical results. We also discuss extensions such as non-autonomous higher-order systems in the appendix.

## 2 First Degree Higher Order ODEs

Let $\overset{n}{x}_j(t)$ be the $n$th time derivative. We consider the system with the system with $J \in \mathbb{N}^+$ variables defined as

$$\mathcal{F}(x_j(t), \dot{x}_j(t), \ldots, \overset{n}{x}_j(t)) = f_j(\boldsymbol{x}(t)), \forall j = 1, \ldots, J \tag{1}$$

More specifically $\mathcal{F}(x_j(t), \dot{x}_j(t), \ldots, \overset{n}{x}_j(t)) = \sum_{i=0}^n \alpha_i \overset{i}{x}_j(t)$ where $\alpha_i \in \mathbb{R}$ and $\overset{0}{x}_j(t) = x_j(t)$. The functions $f_j : \mathbb{R}^J \to \mathbb{R}$ will sometimes be directly referred to as the ODEs. We also emphasize that $f_j$ could be non - linear. We denote $T \in \mathbb{R}^+$ as the maximum time horizon we have the data for, and the trajectory $x_j : [0, T] \to \mathbb{R}$ is a function of time, whereas the state $x_j(t) \in \mathbb{R}, \forall t \in [0, T]$ is a point on the trajectory. We denote the state vector $:= [x_1(t), \ldots, x_J(t)]^\top \in \mathbb{R}^J$ and the vector-valued trajectory function $\boldsymbol{x} := [x_1, \ldots, x_J]$. Now let $f_j^*$'s be the true but unknown ODEs to be uncovered, and $\boldsymbol{x}_i : [0, T] \to \mathbb{R}^j, i \leq N, N \in \mathbb{N}^+$ be the true trajectories that satisfy $f_j^*$'s. In practice, we only measure the true trajectories at discrete times and with noise. Denote the measurement of trajectory $i$ at time $t$ as $\boldsymbol{y}_i(t) \in \mathbb{R}^J$; we assume $\boldsymbol{y}_i(t) = \boldsymbol{x}_i(t) + \epsilon_i(t), \quad \forall i \leq$

---

[*]Webpage: https://www.robertj1.com/. Code: https://github.com/Robertboy18/EDCDE-ICLR-2023

$N, t \in \mathcal{T}$ where $\epsilon_i(t) \in \mathbb{R}^J$ is zero-mean noise with standard deviation $\sigma$. The measurements are made at time $t \in \mathcal{T} = \{t_1, t_2, \ldots, T\}$. We denote the dataset as $\mathcal{D} = \{\boldsymbol{y}_i(t) \mid i \leq N, t \in \mathcal{T}\}$.

## 2.1 GENERALIZED VARIATIONAL FORMULATION

The variational formulation provides a direct link between the trajectory $\boldsymbol{x}$ and the ODE $f_j$ without involving any of the nth derivatives $\overset{n}{x}_j(t)$. In particular consider $J \in \mathbb{N}^+, T \in \mathbb{R}^+$, continuous functions $\boldsymbol{x} : [0, T] \to \mathbb{R}^J, f : \mathbb{R}^J \to \mathbb{R}$, and $g \in \mathcal{C}^n[0, T]$, where $\mathcal{C}^n$ is the set of $n$ times continuously differentiable functions. We define the functionals

$$C_j(f, \boldsymbol{x}, g^n) := \int_0^T \left( f(\boldsymbol{x}(t)) g(t)^n + x_j(t) \Phi(\overset{n}{\dot{g}}(t)) \right) dt, \quad \forall j \in \{1, 2, \ldots, J\} \tag{2}$$

where $\Phi(\overset{n}{\dot{g}})$ is a function of derivatives of $g$. To get the explicit representation of $\Phi$, we must do a Taylor series expansion of $g^n$ and choose the *n-th* term in the series expansion for the corresponding $\overset{n}{\dot{g}}$. We obtain a linear combination of the $\dot{g}$ derivatives. The functional $C_j$ depends now on the testing function $g(t)$ and its $n$ - derivatives $\overset{n}{\dot{g}}(t)$ but not on any $n$ - derivatives of $\overset{n}{x}_j(t)$. Then we can easily consider an extension of the proposition in the paper, which can be stated as follows.

**Proposition 1:** Consider $J \in \mathbb{N}^+, T \in \mathbb{R}^+$, a continuously differentiable function $\boldsymbol{x} : [0, T] \to \mathbb{R}^J$, and continuous functions $f_j : \mathbb{R}^J \times [0, T] \to \mathbb{R}$ for $j = 1, \ldots, J$. Then $\boldsymbol{x}$ is the solution to the system of ODEs in Equation 2 if and only if

$$C_j(f_j, \boldsymbol{x}, g^n) = 0, \forall g \in \mathcal{C}^n[0, T], g(0) = g(T) \equiv 0$$

In regard to the choice of the testing functions, we can consider $g_s(t) = \sqrt{\frac{2}{T}} \sin(\frac{s\pi t}{T})$, which are an orthonormal basis for $L^2[0, T]$ and satisfy $g(0) = g(T) = 0$. Moreover, their derivatives are easy to compute and have a closed-form analytic solution such as for odd powers; we have $\overset{2k+1}{\dot{g}_s}(t) = \frac{(-1)^k \sqrt{2} \pi^k s^k \cos\left(\frac{\pi s x}{T}\right)}{T^{\frac{2k+1}{2}}}$ and for even powers of $n$ we have $\overset{2k}{\dot{g}_s}(t) = \frac{(-1)^k \sqrt{2} \pi^k s^k \sin\left(\frac{\pi s x}{T}\right)}{T^{\frac{2k+1}{2}}}$. The proof of proposition 1 and pseudocode is in A.1 and A.2, respectively.

## 3 NUMERICAL RESULTS

We create generalized ODEs from the ODEs considered in the original paper. In particular, we consider the third-order generalized ODEs (Logistic and Gompertz) given by

$$\overset{3}{x}(t) = -\theta_1 x(t) \cdot \log\left(\theta_2 x(t)\right), \quad \overset{3}{x}(t) = \theta_1 x(t) \cdot \left(1 - x(t)^{\theta_2}\right), \quad \theta_1, \theta_2 \in \mathbb{R}^+ \tag{3}$$

We have no benchmark to compare our EDCDE algorithm as it is difficult to estimate the $\overset{3}{x}$ directly and hence will present the closed-form ODEs discovered by our algorithm. Tables A.3 showcase our results and some of the closed-form ODEs discovered by EDCDE. We fix the number of basis as 50, do 10 simulations for each experiment and set the noise level to 0 for simplicity.

## 4 APPLICATIONS

Similar to the original D-CODE paper, we demonstrate that our generalized algorithms can be used to discover higher-order closed-form ODEs. These algorithms have applications in various fields, such as: **1) Data-Centric AI:** With the increasing availability of data, ML has enabled the construction of more powerful and accurate models. Symbolic regression can be used to analyze complex datasets and identify underlying ODE/PDE systems and relationships that may not be apparent to human analysts. **2) Theory discovery:** Closed-form differential equations provide a concise and interpretable representation of a system's behaviour, making them useful for making predictions and understanding underlying relationships. Symbolic regression can be employed to develop new theories about the system and make predictions about its behaviour.

URM STATEMENT

The authors acknowledge that the author of this work meets the URM criteria of ICLR 2023 Tiny Papers Track.

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

## A  APPENDIX

### A.1  PROOF OF PROPOSITION 1

**Lemma 1:** If we had an ODE of order n, i.e. $\overset{n}{\dot{x}}_1$ then the following statements are true

**1:** $\overset{n}{\dot{x}}_1$ the corresponding Taylor series expansion should be computed until the $n+1$ term, and the first $n$ terms correspond to the boundary terms, which equal to 0 due to $g(0) = g(T) = 0$. The $n+1$ term is the corresponding integral term.

**2:** $\overset{n-1}{\dot{x}}_1$ the corresponding boundary terms equal to the first $n-1$ terms and the $n$ term is the integral. Continuing the process, we eventually get that for the $\dot{x}_1$ term, our boundary term equals the first term, and the integral equals the second term.

**Proof 1:** We now prove our proposition in full generality. As in the original paper, the proof is straightforward, with repeated integration by parts. We have from 2 that

$$\mathcal{F}(x_j(t), \dot{x}_j(t), \dots, \overset{n}{\dot{x}}_j(t)) = f_j(\boldsymbol{x}(t))$$

Now using the Fundamental lemma of the calculus of variations, we get the following two statements are equivalent.

$$\mathcal{F}(x_j(t), \dot{x}_j(t), \dots, \overset{n}{\dot{x}}_j(t)) - f_j(\boldsymbol{x}(t)) = 0 \;\forall t \in [0,T]$$

$$\int_0^T \left( \mathcal{F}(x_j(t), \dot{x}_j(t), \dots, \overset{n}{\dot{x}}_j(t)) - f_j(\boldsymbol{x}(t)) \right) g(t)dt = 0, \quad \forall g \in \mathcal{C}^n[0,T], g(0) = g(T) \equiv 0$$

Using linearity of integration, we get

$$= \int_0^T \left( \mathcal{F}(x_j(t), \dots, \overset{n}{\dot{x}}_j(t)) \right) g(t)dt - \int_0^T f_j(\boldsymbol{x}(t))g(t)dt$$

$$= \int_0^T \left( \sum_{i=0}^n \alpha_i \overset{i}{\dot{x}}_j(t) \right) g(t)dt - \int_0^T \left( f_j(\boldsymbol{x}(t)) \right) g(t)dt$$

$$= \sum_{i=0}^n \int_0^T \alpha_i \overset{i}{\dot{x}}_j(t)g(t)dt - \int_0^T \left( f_j(\boldsymbol{x}(t)) \right) g(t)dt$$

Where the swap can be made due to the dominated convergence theorem (Again, we only have a finite sum, so it is valid mathematically). Now we only care about the first term above. Therefore we get (Avoiding the explicit notation of $t$)

$$= \sum_{i=0}^n \int_0^T \alpha_i \overset{i}{\dot{x}}_j(t)g(t)dt = \sum_{i=0}^n \left( \alpha_i \left( \sum_{k=1}^i \xi_k\{g^n(t)\} \overset{k}{\dot{x}} - \int_0^T \xi_{i+1}\{g^n(t)\}x_j dt \right) \right)$$

$$= \sum_{i=0}^n \left( -\int_0^T \xi_{i+1}\{g^n(t)\}x_j dt \right)$$

where in the last equality we used the fact that $g(0) = g(T) = 0$ and where $\xi_k\{g^n(t)$ is the $k - th$ term in the power series expansion of $g^n(t)$. That means that

$$= -\int_0^T \sum_{i=0}^n \xi_{i+1}\{g^n(t)x_j\}dt = \int_0^T \Phi(\overset{n}{\ddot{g}}(t))x_j dt = \int_0^T \left( f(\boldsymbol{x}(t))g(t)^n + x_j(t)\Phi(\overset{n}{\ddot{g}}(t)) \right) dt$$

This proves that (Let $\mathcal{J} = \{1, 2 \ldots J\}$ and $\forall j \in \mathcal{J}$)

$$\mathcal{F}(x_j(t), \dot{x}_j(t), \ldots, \overset{n}{\ddot{x}}_j(t)) = f_j(\boldsymbol{x}(t)) \iff C_j(f_j, \boldsymbol{x}, g^n) = 0, \ \forall g \in \mathcal{C}^n[0,T], \ g(0) = g(T) = 0$$

## A.2 PSEUDOCODE

---
**Algorithm 1** EDCDE - ODE
---
   **Input:** Dataset $\mathcal{D} = \{\boldsymbol{G}_i(t) \mid i \le N, t \in \mathcal{T}\}$
   **Input:** Smoothing Algo $S$, Optimization Algo $O$
   **Input:** Numerical Integration Algo $I$, Order $n$
   **Input:** Test functions $g_s, s \le S$, Initial guesses $f_j, j \le J$ $\widehat{x}_i = \mathcal{S}(\boldsymbol{x}_i(t_1), \boldsymbol{x}_i(t_2), \ldots), i \le N$
   $\Phi$ = Symbolically differentiate $g(x)^n$
   **for** $i = 1$ **to** $J$ **do**
      $\hat{f}_j = f_j$
      Converge = False
      **while** Not Converge **do**
         obj = $\sum_{i=1}^N \sum_{s=1}^S C_j(f_j, \widehat{x}_i, g_s^n)$ [Use $\Phi$]
         $\hat{f}_j$, Converge = $\mathcal{O}(obj, \hat{f}_j)$
      **end while**
   **end for**
   **Output:** The discovered ODE's $f_j, \ j \le J$
---

## A.3 NUMERICAL RESULTS

We note that we did not compare it with the original DCODE algorithm as DCODE does not estimate $\overset{3}{\dot{x}}(t)$ directly, but instead, they convert that into a first-order autonomous system because higher order or nonautonomous ODEs can be transformed into first-order autonomous systems by including more variables. We also did not compare to the original D-CODE paper as the original codebase did not include examples on how to do this and estimate higher-order ODEs.

| Higher Order ODE Training Time | | |
| --- | --- | --- |
| ODE | Training Time | Order |
| Logistic | 1540 | 1 |
| Gompertz | 1200 | 1 |
| Logistic | 1080 | 3 |
| Gompertz | 3000 | 3 |

| Higher Order ODE Closed Form Discovery | | |
| --- | --- | --- |
| ODE | Order | ODE Discovered (Best) |
| Logistic | 1 | $\theta_1 x(t) - x(t)^{\theta_2}$ |
| Gompertz | 1 | $-\theta_1 x(t) \cdot \log(\theta_2 x(t))$ |
| Logistic | 3 | $x(t)^{\theta_1}(\theta_2 - x(t))$ |
| Gompertz | 3 | $(-x(t) - 4\log x(t))(\theta_1 \log(\theta_2 x(t)))$ |

## A.4 COMPUTATIONAL COST AND THE TEST FUNCTION

The algorithm for D-CODE remains almost the same as the original paper. Still, we can improve the complexity of the original algorithm by pre-computing all the derivatives up to order $n$ and then passing it onto our objective function and optimization step. A key benefit to this is that our test function has a very nice symmetry in the derivatives, i.e. differentiating $\sin(x)$ twice gives us back the original function (the negative of it). This symmetry allows us to use the generalized derivatives formula for odd and even derivative powers of $g_s(t)$ by plugging in the value for $k$. We can also do the same for $\alpha_i$ in section D by pre-computing and storing all the derivatives. However, one must remember that their closed-form derivatives could be complicated to compute, especially for higher orders. In theory, our algorithm should work and be generalizable to capture most of the ODEs, but in practice, rarely does one encounter greater than third-order ODEs, so our method is feasible.

## B  CONVERGENCE TO DISTANCE

The convergence properties remain the same as in the original paper, and the proof of the theorem follows the same procedure as the original paper but replaces $g_s$ by $g_s^n$.

**Theorem 2:** Consider $J \in \mathbb{N}^+, j \in \{1, \ldots, J\}, T \in \mathbb{R}^+$. Let $f^* : \mathbb{R}^J \to \mathbb{R}$ be a continuous function, and let $\boldsymbol{x} : [0, T] \to \mathbb{R}^J$ be a continuously differentiable function satisfying $\dot{x}_j(t) = f^*(\boldsymbol{x}(t))$. Consider a sequence of functions $(\widehat{\boldsymbol{x}}_k)$, where $\widehat{\boldsymbol{x}}_k : [0, T] \to \mathbb{R}^J$ is a continuously differentiable function. If $(\widehat{\boldsymbol{x}}_k)$ converges to $\boldsymbol{x}$ in $L^2$ norm. Then for any Lipschitz continuous function $f$

$$\lim_{S \to \infty} \lim_{k \to \infty} \sum_{s=1}^{S} C_j\left(f, \widehat{\boldsymbol{x}}_k, g_s^n\right)^2 = d_{\boldsymbol{x}}\left(f, f^*\right)^2, \text{ where } \{g_1, g_2, \ldots\} \text{ is a Hilbert (orthonormal) basis}$$

for $L^2[0, T]$ such that $\forall i, g_i(0) = g_i(T) = 0$ and $g_i \in \mathcal{C}^n[0, T]$.

## C  NON - AUTONOMOUS ODES

We consider a Non -Autonomous system of ODEs as $\mathcal{F}(x_j(t), \dot{x}_j(t), \ldots, \overset{n}{\dot{x}}_j(t)) = f_j(\boldsymbol{x}(t), t), \forall j = 1, \ldots, J$. The only key difference here is to make one variable substitution, i.e. $x_m = t$, and so $\dot{x}_m = 1$, and we solve the following ODE system

$$\mathcal{F}(x_j(t), \dot{x}_j(t), \ldots, \overset{n}{\dot{x}}_j(t)) = f_j(\boldsymbol{x}(t), t) \text{ and } \dot{x}_m = 1, \ m = J + 1, \quad \forall j = 1, \ldots, J \quad (4)$$

Then we can easily use our generalized D-CODE algorithm given the measurements of $x_m$ and $x_j$.

## D  FIRST DEGREE GENERAL ODES

Lastly, we can generalize this technique to discover more general ODE systems that are of the form

$$\mathcal{F}(x_j(t), \dot{x}_j(t), \ldots, \overset{n}{\dot{x}}_j(t)) = f_j(\boldsymbol{x}(t)), \forall j = 1, \ldots, J \quad (5)$$

More specifically $\mathcal{F}(x_j(t), \ldots, \overset{n}{\dot{x}}_j(t)) = \sum_{i=0}^{n} \alpha_i(x_j) \overset{i}{\dot{x}}_j(t)$ where $\alpha_i$ are now functions of $x_j$. This is a more general setting, and we can again use our generalized variational formulation to uncover the true $f_j$. This only works assuming all our $\alpha_i(x_j(t)) \in C^n[0, T]$. We come to the following proposition

**Proposition 2:** Consider $J \in \mathbb{N}^+, T \in \mathbb{R}^+$, continuous functions $\boldsymbol{x} : [0, T] \to \mathbb{R}^J, f : \mathbb{R}^J \to \mathbb{R}$, and $\alpha_i, g \in \mathcal{C}^n[0, T]$, where $\mathcal{C}^n$ is the set of $n$ times continuously differentiable functions. We define the functionals

$$C_j(f, \boldsymbol{x}, g^n) = \int_0^T (f(\boldsymbol{x}(t))g(t)^n + x_j(t)\Phi(\overset{n}{\dot{g}}(t), \overset{n}{\dot{\alpha}}_i))dt \quad \forall j \in \{1, 2, \ldots, J\}$$

and where $\Phi(\overset{n}{\dot{g}}(t), \overset{n}{\dot{\alpha}}_i)$ is a function of derivatives of $g$ and $\alpha_i$. We can again prove the proposition in full generality as it follows identical steps as the previous proposition, but now, considering the influence of $\alpha_i$ and their derivatives. We leave that proof of this for future work.

