# OpenReview forum: "EDCDE -  Extended Discovery of Closed-Form Differential Equations"
_ICLR.cc/2023/TinyPapers — Submitted to Tiny Papers @ ICLR 2023_

### Official Review · Reviewer_WNj5 · 2023-03-20

**Confidence:** 2

**Summary Of Contributions:**

This tiny paper extends a previous work of symbolic regression for discovering ODE (D-CODE) to a setting which works for discovering higher order ODEs. Proofs and numerical results were provided to illustrate the application of the proposed method.

**Rating:**

Clear, Correct, and Reproducible (CCR): a submission which meets the reviewing criteria

**Strengths And Weaknesses:**

Strengths:
1. The paper is well-written, concise and has a strong motivation for the work presented.
2. Numerical results are provided along with proofs of the proposed method
3. The paper also mets the formatting requirements.

Weaknesses:
1. While I appreciate the conciseness, some additional references and introduction would have improved the paper for readers not familiar with the previous work.
2. Additional results comparing the proposed method to the previous method would have made the results more compelling.
3. Some typos

**Suggested Changes:**

1. There seems to be some formatting and typos in the paper. For example, the first sentence in Section 2 is not capitalized and potentially had repeated phrases. Section 3. also refers to Table A3 twice.

2. I understand that in general, it might be challenging to provide comparisons as a benchmark to the proposed algorithm. However, it would make the lack of comparisons more compelling if the authors could explain why their method couldn't at least be benchmarked against the original method (D-CODE) in terms of the timing and the ODE discovered?

3. I believe that the paper's readability could be improved if the authors could also cite several seminal papers relevant to the field, in case the readers are not familiar and would like to get a better background on the work.

All the best!

---

> ### Author Response · Authors · 2023-05-05
> **Official Comment by Author**
>
> Thank you for your feedback. We appreciate your careful review of our paper. We will address your concerns as follows, point by point. We agree that this will help improve our paper's clarity and quality.
>
> ## Reference section
> As mentioned in the other comment, we acknowledge that there is limited space in the paper to include a detailed literature review, but we have made an effort to include a brief background and relevant references in the introduction section.
>
> We would also like to point out that our proposed method is an extension of the D-CODE paper, which is the main reference for our work. However, we understand the importance of discussing related works and their differences and similarities to our proposed method. We cannot do a full-fledged review as there is no space (it could take around 2 additional pages), but we refer the readers to check the original paper as they include a detailed introduction to the field. Therefore, we have revised the introduction section to include more references and provide a better context for our approach.
>
> However, if the reviewer feels it would still be important, we will include it in the next revised version (if need be).
>
> ## Lack of benchmarks
> Again as mentioned in the other review, this paper is just an extension theoretically and more experiments in the future will be done. We further point out that our proposed method is an extension of the D-CODE paper, which only has results for first-order ODEs. Therefore, our examples in the numerical experiments are intended to demonstrate the capabilities and robustness of our extended approach rather than to provide a baseline for comparison.
>
> ## Other minor changes
> All those issues have been fixed too.
>
> Thank you again for your thoughtful comments. If you have any additional questions or concerns, please do not hesitate to let us know. The new revised version of the paper will also be uploaded by today.

---

### Official Review · Reviewer_vo5R · 2023-04-02

**Confidence:** 4

**Summary Of Contributions:**

The paper proposes an extension of an older ICLR 2022 article “D-CODE: Discovering closed form ODEs from closed trajectories”, that proposes a symbolic regression framework that approximates  time derivatives from time-series data variationally. The present paper proposes an extension of the framework to systems with higher order derivatives by considering higher order derivatives of the test functions employed in the variational formulation.

**Rating:**

Needs Clarification (NC): a submission which does not meet the reviewing criteria and needs clarification for its described problem or solution

**Strengths And Weaknesses:**

strengths:
+ Interesting problem, that might be nevertheless challenging to solve adequately due to the difficulty to estimate the order of the observed system (to my understanding).

+ Clearly formulated aim of the project.


weaknesses:
- Several typos (e.g., section 2 first sentence)

- Inconsistencies between proposed formalism and claims/numerical experiments (see suggested changes)

- No literature review


**Suggested Changes:**

-I am a little bit confused with the assumptions of the method, so I think they require some clarification. In particular, in section 2 third line, the authors mention that $\mathcal{F}$ (the observed system) is a linear combination of up to n-th derivatives of $x(t)$. What I understand from this is assumed that the observed system is a linear system of n-th order. However the numerical experiments of the proposed method are applied to nonlinear systems. Here it needs some clarification if the assumption about linearity of $\mathcal{F}$ was indeed what the authors assume, and if the numerical experiment on purpose deviates from their assumption to test the capabilities and robustness of the approach. If yes I would think that another simpler example that follows the initial assumptions is also required to establish some baseline performance. Otherwise one of these two points have to be changed to resolve the inconsistency.

-No discussion or reference to previous literature excluding D-CODE paper. However there is limited space to do so, but I would suggest adding a section in an appendix.

-Not clear to me how one can know/infer the order of the observed system from the observations. Or is there the assumption that if one includes more orders in the estimation, the corresponding terms will be identified as almost zero?

-Line 9 section 2: “as in the paper”. You might want to mention the citation here.

-Typos at the beginning of section 2.

---

> ### Author Response · Authors · 2023-05-05
> **Official Comment by Author**
>
> Thank you for your feedback. We appreciate your careful review of our paper. We will address your concerns as follows point by point. We agree that this will help improve our paper's clarity and quality.
>
> ## Linearity of the system
>
> We apologize for any confusion caused by our phrasing in the paper. To clarify, in our proposed method, we assume that the left-hand side of the equation is linear in terms of the derivatives, meaning that we do not have any higher degree terms in an ODE. However, we do allow for the right-hand side function f to be non-linear, which is what we aim to uncover using our method.
>
> Regarding the numerical experiments, we understand your suggestion that another simpler example that follows the initial assumptions is required to establish some baseline performance. This paper is just an extension theoretically, and more experiments in the future will be done. However, we would like to point out that our proposed method is an extension of the D-CODE paper, which only has results for first-order ODEs. Therefore, our examples in the numerical experiments are intended to demonstrate the capabilities and robustness of our extended approach rather than to provide a baseline for comparison. Hope this clarifies any confusion about our assumptions and experimental design.
>
> ## Reference section
>
> We acknowledge that there is limited space in the paper to include a detailed literature review, but we have made an effort to include a brief background and relevant references in the introduction section.
>
> We would also like to point out that our proposed method is an extension of the D-CODE paper, which is the main reference for our work. However, we understand the importance of discussing related works and their differences and similarities to our proposed method. Therefore, we have revised the introduction section to include more references and provide a better context for our approach.
>
> ## Infer the order of the system
>
> You are correct that it may not be possible to directly infer the order of the observed system from the observations, and it is up to the domain expert to set the order a priori. As stated in the D-CODE paper, the success of ODE discovery depends on the availability of a well-curated dataset, particularly in terms of variable selection. In particular, "The domain expert should decide which variables to include in the dataset based on their knowledge of the problem being addressed and the interactions between variables. The iterative nature of scientific discovery is also emphasized in the D-CODE paper, which involves hypothesis generation, data curation, modelling, validation, and so on. D-CODE focuses on the modelling component, but it can facilitate the entire loop of discovery since it distills an interpretable closed-form ODE."
>
> Regarding the estimation of the order of the system, it is assumed in the proposed method that the observed system is a linear combination of up to n-th derivatives of a function $f$, and that the method estimates the coefficients of these derivatives by solving an optimization problem. The proposed method does not explicitly assume that higher-order terms will be identified as almost zero if they are not present in the system.
>
> ## Other minor changes
> All of them have been corrected now.
>
> Thank you again for your thoughtful comments. If you have any additional questions or concerns, please do not hesitate to let us know. The new revised version of the paper will also be uploaded by today.

---

### Author Response · Authors · 2023-05-30
**Archival Information**

Thank you for your time and effort in reviewing my paper. I greatly appreciate your valuable feedback and suggestions for improvement. I have opted for my paper to be considered for archival.

---

### Meta-Review · Area_Chair_B8NB · 2023-04-04

**Recommendation:** Invite to revise
**Confidence:** 3

**Metareview:**

This concise paper tackles an interesting problem. While minor errors and inconsistencies present in the manuscript, it provides both theoretical arguments and experimental results to demonstrate the effectiveness of the proposed framework.

**Summary:**

The paper extends the popular D-CODE method by intruding a generalized variational framework.

**Reason For Not Giving A Higher Recommendation:**

The theoretical arguments and numerical evidence are not very well structured. More relevant literature needs to be discussed.

**Reason For Not Giving A Lower Recommendation:**

The paper is well written.

---

> ### Author Response · Authors · 2023-05-05
> **Official Comment by Author**
>
> We thank the meta-reviewers for their decision and the points made. We have addressed all the concerns and have changed the introduction to add more relevant literature. We again emphasize this paper was more of a theoretical extension of the original proposed variational formulation. Therefore, our examples in the numerical experiments are intended to demonstrate the capabilities and robustness of our extended approach rather than to provide a baseline for comparison. We hope this clarifies any confusion about our assumptions and experimental design.
>
> We will ensure to incorporate and clarify all changes made by the reviewers. Thank you for the decision once again.

---

### Decision · Program_Chairs · 2023-04-10

Revision accepted; invite to archive